# Diagnostic Accuracy of Line-Field Confocal Optical Coherence Tomography for the Diagnosis of Skin Carcinomas

**DOI:** 10.3390/diagnostics13030361

**Published:** 2023-01-18

**Authors:** Elisa Cinotti, Tullio Brunetti, Alessandra Cartocci, Linda Tognetti, Mariano Suppa, Josep Malvehy, Javiera Perez-Anker, Susanna Puig, Jean Luc Perrot, Pietro Rubegni

**Affiliations:** 1Department of Medical, Surgical and Neurological Sciences, Dermatology Section, University of Siena, 53100 Siena, Italy; 2Groupe d’Imagerie Cutanée Non Invasive (GICNI), Société Française de Dermatologie (SFD), 75008 Paris, France; 3Department of Dermatology, Hôpital Erasme, Université Libre de Bruxelles, 1050 Brussels, Belgium; 4Melanoma Unit, Hospital Clinic Barcelona, University of Barcelona, 08007 Barcelona, Spain; 5CIBER de Enfermedades Raras, Instituto de Salud Carlos III, 08007 Barcelona, Spain; 6Department of Dermatology, University Hospital of St-Etienne, 42270 Saint-Etienne, France

**Keywords:** optical coherence tomography, tumor, basal cell carcinoma, imaging, squamous cell carcinoma

## Abstract

Line-field confocal optical coherence tomography (LC-OCT) is a new, noninvasive imaging technique for the diagnosis of skin cancers. A total of 243 benign (54%) and malignant (46%) skin lesions were consecutively enrolled from 27 August 2020, to 6 October 2021 at the Dermatology Department of the University Hospital of Siena, Italy. Dermoscopic- and LC-OCT-based diagnoses were given by an expert dermatologist and compared with the ground truth. Considering all types of malignant skin tumours (79 basal cell carcinomas (BCCs), 22 squamous cell carcinomas, and 10 melanomas), a statistically significant increase (*p* = 0.013) in specificity was observed from dermoscopy (0.73, CI 0.64–0.81) to LC-OCT (0.87, CI 0.79–0.93) while sensitivity was the same with the two imaging techniques (0.95 CI 0.89–0.98 for dermoscopy and 0.95 CI 0.90–0.99 for LC-OCT). The increase in specificity was mainly driven by the ability of LC-OCT to differentiate BCCs from other diagnoses. In conclusion, our real-life study showed that LC-OCT can play an important role in helping the noninvasive diagnosis of malignant skin neoplasms and especially of BCCs. LC-OCT could be positioned after the dermoscopic examination, to spare useless biopsy of benign lesions without decreasing sensitivity.

## 1. Introduction

Line-field confocal optical coherence tomography (LC-OCT) is a new, noninvasive skin imaging technique that combines the advantages of optical coherence tomography (OCT) and reflectance confocal microscopy (RCM) in terms of spatial resolution, penetration, and image orientation, overcoming their respective limits [1,2,3,4,5]. LC-OCT has a higher resolution than OCT (~1 μm) [6,7,8] and higher penetration [9] depth than RCM (~500 μm), and it creates both vertical and horizontal images in real time [10,11,12,13].

Recently, LC-OCT [14] has been gaining attention because it has been shown that this device can help the clinical diagnosis of different neoplastic [15,16,17,18], inflammatory [19,20,21,22,23], and infectious [24,25] skin diseases. In particular, it has proven to be very effective in identifying basal cell carcinoma (BCC) [26], even managing to differentiate its histological subtypes [27] and to follow-up noninvasive treatment [28]. Furthermore, this noninvasive diagnostic technique can be used to help the differentiation of actinic keratosis (AK) from squamous cell carcinoma (SCC) [29,30,31] and to monitor the field of cancerization-directed treatments [32].

Many descriptive studies have shown the relevance of this imaging device for the diagnosis of cutaneous tumours, and our study aimed to evaluate the sensitivity and specificity of LC-OCT compared to dermoscopy for the diagnosis of skin tumours in a real-life setting in a third-level dermatology department. 

## 2. Materials and Methods

### 2.1. Study Design

Prospective observational, monocentric study.

### 2.2. Setting

Patients were enrolled from the 27 August 2020 to the 6 October 2021 at the Dermatology Department of the University Hospital of Siena, Italy, from the melanoma prevention outpatient ambulatory. The study was conducted according to the criteria set by the Declaration of Helsinki. All data were deidentified before use. 

### 2.3. Participants 

We enrolled consecutive patients with cutaneous lesions of clinical and/or dermoscopic uncertain diagnosis of possible malignant skin tumours that needed to be removed or followed up according to a skin imaging expert dermatologist (E.C.) and that had LC-OCT examination.

### 2.4. Imaging Examination

Dermoscopy was performed both with a hand-held 10× dermoscope (DermLite DL4, DermLite, San Juan Capistrano, USA) and a 20× videodermoscope (Vivacam, Mavig, Munich, Germany). LC-OCT (DeepLive, Damae, France) was performed in horizontal and vertical scans and 3D mode, and a video was acquired on the operator’s judgment. Following manufacturer recommendations, lesions in the patients’ periocular region were excluded from LC-OCT examination. 

Dermoscopic and LC-OCT diagnoses were given by an expert in skin imaging (E.C.) during the imaging examination of the lesions and were registered on the LC-OCT device. Concerning LC-OCT, BCC was diagnosed by the presence of tumour lobules [27], SCC by the presence of atypical keratinocytes in the entire epidermis [31,32], and melanoma by the presence of atypical bright cells that were sparse inside the epidermis and/or inside melanocytic nests [12,13,14,15]. The lesions suggesting malignant skin tumours at dermoscopic and/or LC-OCT examination were biopsied or surgically removed for histological diagnosis. The others were followed up for at least one year. 

### 2.5. Statistical Analysis

The sensitivity and specificity of each technique (LC-OCT and dermoscopy) for the diagnosis of BCC, SCC/Bowen disease (i.e., SCC or Bowen disease), AK/SCC/Bowen disease (i.e., AK or SCC or Bowen disease) group and malignant tumour were calculated with their exact 95% confidence interval (CI) by using the histopathological diagnosis obtained from an incisional or excisional biopsy as the gold standard; a distinct analysis was also performed, including the lesions that had a final diagnosis based on a follow up of at least one year without a histological examination. 

The sensitivity and specificity of LC-OCT and dermoscopy for the diagnosis of the different skin tumours were compared by using the proportion test. *p* values < 0.05 were considered statistically significant. All statistical analyses were conducted by using R (version 4.0.3., R foundation for statistical computing). 

## 3. Results

We included 196 patients (81 women, 115 men; mean age of 64.45 years, range 0–96 years) with 243 lesions; 226 lesions were histopathologically confirmed (Table 1) and 17 lesions had a final diagnosis after a follow-up of at least one year. Sensitivity and specificity of dermoscopy and LC-OCT for BCC, SCC/Bowen disease group, AK/SCC/Bowen disease group, and malignant tumour considering histopathology as the gold standard are reported in Table 2.

### 3.1. Diagnostic Performances of Dermoscopy and LC-OCT Considering Only Cases with Histological Diagnoses

#### 3.1.1. Dermoscopy and LC-OCT Diagnostic Performances for BCC

Considering the 79 histopathologically confirmed BCCs, LC-OCT showed higher specificity (0.95, CI 0.90–0.98; *p* = 0.015) for BCC diagnosis than dermoscopy (0.86, CI 0.80–0.91, *p* = 0.015), and no statistically significant difference in sensitivity (0.97 CI 0.91–1.00 for LC-OCT and 0.96 CI 0.89–0.99 for dermoscopy; Table 2). Dermoscopy had 20 false positive (FP) cases that histologically corresponded to two nevi, one AK, two Bowen diseases in situ, three inflammatory lesions, a scar, and 11 benign nonmelanocytic lesions (including a solar lentigo, a seborrheic keratosis (SK), a lichenoid keratosis, a xanthogranuloma, a sebaceoma, a trichilemmoma, a trichoblastoma, and a neurofibroma). LC-OCT had seven FP cases that histologically corresponded to one nevus, one inflammatory lesion, and five benign nonmelanocytic lesions. LC-OCT enabled us to correctly diagnose 13 of the 20 dermoscopic FP BCCs as four inflammatory lesions, two cases of normal skin, two Bowen diseases, one AK, one nevus, one neurofibroma, one scar, and one xanthogranuloma. Three dermoscopic false negative (FN) cases were diagnosed at dermoscopy as AK, SCC, and SK and histologically corresponded to two infiltrative BCCs and a superficial microinvasive BCC. Two of these FN cases were also FN at LC-OCT and were diagnosed as AK and SCC, while they corresponded to two infiltrative BCCs at histopathology.

#### 3.1.2. Dermoscopy and LC-OCT Diagnostic Performances for the Diagnosis of SCC/Bowen Disease

LC-OCT showed a slightly higher sensitivity (0.86, CI 0.65–0.97) for SCC/Bowen disease diagnosis than dermoscopy (0.77, CI 0.55–0.92), which did not reach a statistically significant difference. Concerning specificity, no statistically significant difference was found (0.98, CI 0.94–0.99 for LC-OCT and 0.96,0.92–0.98 for dermoscopy; Table 2). Among the 22 histopathologically confirmed SCC/Bowen diseases, there were nine FP at dermoscopy, five FP at LC-OCT, five FN on dermoscopy, and three FN on LC-OCT (two of them were Bowen diseases in situ diagnosed as AK). The nine FP at dermoscopy were histologically diagnosed as one BCC, three AKs, two inflammatory lesions, one dermatofibroma, one granulomatous lesion, and one microcystic adnexal carcinoma; the five FP at LC-OCT were histologically diagnosed as one BCC, one AK, one inflammatory lesion, one dermatofibroma, and one granulomatous lesion.

The five FN at dermoscopy were diagnosed as one AK, two BCCs, one SK, and one granuloma, and corresponded to four Bowen diseases in situ and one microinvasive keratoacanthoma at histopathology; the three FN at LC-OCT were diagnosed as two AKs and one SK and corresponded to two Bowen diseases in situ and a microinvasive keratoacanthoma at histopathology. 

#### 3.1.3. Dermoscopy and LC-OCT Diagnostic Performances for the Diagnosis of AK/SCC/Bowen Disease

LC-OCT showed higher sensitivity and specificity than dermoscopy for AK/SCC/Bowen disease diagnosis (sensitivity of 0.87 (CI 0.72–0.96) for dermoscopy and 0.95 (CI 0.82–0.99) for LC-OCT, specificity of 0.96 (CI 0.92–0.98) for dermoscopy and 0.97 (CI 0.93–0.99) for LC-OCT (Table 2)). However, the difference in sensitivity and specificity did not reach statistical significance.

#### 3.1.4. Dermoscopy and LC-OCT Diagnostic Performances for Malignant Tumour

Considering only the cases with a histologic diagnosis of malignancy, we observed a significant increase of specificity from 0.73 (CI 0.64–0.81) with dermoscopy to 0.87 (CI 0.79–0.93) with LC-OCT (*p* = 0.013) for a malignant tumour, whereas the sensitivity was similar with the two imaging techniques (0.95 CI 0.89–0.98 for dermoscopy and 0.95 CI 0.90–0.99 for LC-OCT). The group of malignant tumours included both skin carcinomas and melanomas.

#### 3.1.5. Diagnostic Performances of Dermoscopy and LC-OCT Considering Both Histological and Follow-Up Diagnoses

The sensitivity and specificity of dermoscopy and LC-OCT for BCC and malignant tumour considering as comparison the diagnoses derived from histopathology and follow-up at least one year are reported in Table 3.

#### 3.1.6. Dermoscopy and LC-OCT Diagnostic Performances for BCC (Including 13 Cases without a Histological Diagnosis)

Considering both the cases with histology and follow-up of at least one year, LC-OCT enabled us to correctly diagnose 26 over 33 dermoscopic FP cases: seven inflammatory lesions, three cases of normal skin, three scars, two Bowenoid SCCs in situ, two AKs, three nevi, one neurofibroma, one xanthogranuloma, one seborrhoeic keratosis, one rosacea, one sebaceous hyperplasia, and one scaly crust with papillomatosis. Among these cases, LC-OCT allowed us to save 13 excisions. Considering these 13 FP lesions at dermoscopy for which BCC was excluded after LC-OCT and for which surgical excision was not done (assuming that the follow-up >1 year of these patients could confirm the absence of BCC), the specificity for BCC diagnosis increased from 0.79 (CI 0.72–0.85) for dermoscopy to 0.96 (CI 0.91–0.98) for LC-OCT (*p* < 0.001). Sensitivity was similar for LC-OCT (0.97, CI 0.91) and dermoscopy (0.96, CI 0.89–0.99, Table 3).

#### 3.1.7. Dermoscopy and LC-OCT Diagnostic Performances for Malignant Tumours (Including 17 Cases without a Histological Diagnosis)

Considering the 17 FP lesions at dermoscopy for which malignancy was excluded after LC-OCT and surgical excision was not done (assuming that the follow-up of these patients could confirm the absence of malignant tumour), the specificity for malignancy increased respectively from 0.64 (CI 0.55–0.72) of dermoscopy to 0.89 (CI 0.82–0.93, *p* < 0.001) for LC-OCT. Sensitivity was similar: 0.95 (CI 0.90–0.99) for LC-OCT and 0.95 (CI 0.89–0.98) for dermoscopy (Table 3).

## 4. Discussion

Our study showed that LC-OCT can increase specificity for the noninvasive diagnosis of skin cancers compared to dermoscopy. Considering histopathology as a gold standard and analyzing only the cases with histological diagnosis (Table 2), we found an increase in specificity for the diagnosis of BCC from 0.86 (0.80–0.91 CI) for dermoscopy to 0.95 (0.90–0.98 CI) for LC-OCT (*p* = 0.015). The sensitivity was similar with the two methods (0.96 CI 0.89–0.99 for dermoscopy with three FN cases and 0.97 CI 0.91–1.00 for LC-OCT with two FN cases). 

The same analysis including the cases that were diagnosed based on a follow-up of at least one year and that lacked histopathological examination obtained similar results. We found an increase in the specificity for the diagnosis of BCC from 0.79 (CI 0.72–0.85) for dermoscopy to 0.96 (CI 0.91–0.98) for LC-OCT (*p* < 0.001), whereas with regard to sensitivity we did not find any statistically significant difference between dermoscopy (0.96 CI 0.89–0.99) and LC-OCT (0.97 CI 0.91–1.00). Similar sensitivity results probably reflect the current use of LC-OCT as a secondary-level technique on skin lesions that are already identified as suspicious by dermoscopic examination. In most cases, LC-OCT easily confirms the dermoscopic diagnosis of a malignant tumour [27] and it is interesting to note that in clinical practice, LC-OCT is useful to increase the diagnostic confidence of the dermatologist and to confirm the need for surgical excision. 

LC-OCT only missed two infiltrative BCCs, and retrospective examination of their images revealed hyperkeratosis thickness ranging from 200 to 300 µm with no visible dermis in one case and poor image in the other case

Concerning specificity, LC-OCT significantly reduced the cases of false positives (FP) BCCs in our series. 13 FP cases of BCC at dermoscopy were correctly diagnosed with LC-OCT (Table 3; Figure 1 and Figure 2). These data are consistent with the latest studies on LC-OCT that highlight how this technique can easily recognize BCC imitators [20,33,34,35]. 

There were only seven FP cases at LC-OCT, and three of them corresponded to benign skin tumours that can share histopathological similarities with BCC: a sebaceoma, a trichoblastoma, and a trichilemmoma. Sebaceoma (Figure 3) is characterized by multiple basal cell nests (Figure 3, asterisk) with a random mix of sebaceous cells in the upper and middle dermis with possible continuity with the basal layer of the epidermis [36]. Trichoblastoma (Figure 4) shows irregular nests of basal cells similar to BCC, with variable stromal thickening and pilar differentiation [36]. Trichilemmoma (Figure 5) is composed of one or more lobules (Figure 5, asterisk) in the dermis that extend in continuity with the epidermis (Figure 5, orange arrow) or the follicular epithelium, and there is a peripheral layer of columnar palisade cells [36]. Under LC-OCT, all these three tumours exhibited tumour islands with overlapping features of BCC. We should also consider that there could be biopsy sampling errors explaining some FP results. Interestingly, one LC-OCT FP case of our series, defined on the histological report of an incisional biopsy as solar lentigo, was later completely excised based on the retrospective revaluation of the LC-OCT images and had a final histological diagnosis of BCC.

In the literature, we could find only one prospective study on the diagnostic accuracy of LC-OCT for skin tumours, and it consists of similar real-life research on equivocal lesions. It showed that LC-OCT can significantly increase diagnostic confidence after dermoscopy and avoid potentially unnecessary biopsies [37]. However, it revealed a higher sensitivity (0.98) and a good, but lower, specificity (0.80) for LC-OCT compared to dermoscopy (sensitivity of 0.90 and specificity of 0.86).

The acquisition and interpretation of the LC-OCT images are operator-dependent and different results may be related to different investigator expertise and different lesion selection (equivocal aspect of the lesion at dermoscopy). Moreover, Gust et al. found that in 70% of the lesions, the LC-OCT diagnostic was provided with high confidence in comparison with dermoscopy which only provided high confidence in 48% of the lesions [37]. In this subgroup, the LC-OCT performance increased significantly, with a sensitivity of 100% and a specificity of 97% in agreement with our results. In the future, an effort should be made to define precise criteria for the LC-OCT diagnosis of skin tumours to have more reliable and comparable results. Moreover, artificial intelligence could help the identification of BCC tumour lobules and atypical cells [10]. Regarding the diagnosis of SCC/Bowen’s disease/AK, we found an increase in both sensitivity and specificity for LC-OCT compared to dermoscopy without any statistically significant difference. Recently, many studies have shown that LC-OCT can identify several histological criteria of AK and SCC and this technique seems to be promising for the diagnosis of squamous cell tumours [16,29,31,32]. Our study could not prove a statistically significant benefit of LC-OCT possibly due to the relatively small sample size that has been analyzed. 

Regarding the diagnosis of malignancy, considering histopathology as a gold standard and analyzing only cases with histological diagnosis, a significant increase in specificity (Table 2) was observed from 0.73 (IC 0.64–0.81) with dermoscopy to 0.87 (IC 0.79–0.93) with LC-OCT (*p* = 0.013). However, we did not detect any statistically significant difference in sensitivity between the two methods (0.95 IC 0.89–0.98 for dermoscopy and 0.95 IC 0.90–0.99 for LC-OCT), similar to the diagnosis of BCCs. The same analysis including the 17 FP cases of malignity at dermoscopy in which the diagnosis of malignant neoplasm was excluded with LC-OCT and surgery was not performed (assuming that the follow-up of these patients could confirm the absence of malignant tumour, (see Table 3)) obtained similar results. We detected an increase in specificity for malignant skin tumours from 0.64 (CI 0.55–0.72) for dermoscopy to 0.89 (CI 0.82–0.93) for LC-OCT (*p* << 0.001) while regarding the sensitivity we did not find any statistically significant difference between dermoscopy (0.96 CI 0.89–0.99) and LC-OCT (0.97 CI 0.91–1.00). These data on malignant tumours were mainly driven by BCCs and SCCs because melanomas were few. The increase in specificity with LC-OCT for the diagnosis of malignant skin tumours was mainly determined by the increase in specificity for the diagnosis of BCC. Although LC-OCT seems to play a possible role also for melanocytic tumours [12], to date there are few data on the diagnostic accuracy of LC-OCT for malignant skin tumours other than BCC and SCC [15].

## 5. Conclusions

Our real-life study confirmed that dermoscopy can select lesions at risk of being malignant skin tumours (very sensitive tool).LC-OCT could be positioned in a second line to rule out malignancy to spare useless biopsy without decreasing sensitivity (very specific tool).LC-OCT can help in the identification of BCC with only 10 diagnostic errors in our entire database covering more than one year.LC-OCT seems to also be promising for keratinocyte tumours (AK, SCC, and Bowen’s disease) by increasing the specificity and reducing FP cases compared to dermoscopy.Further studies should be performed to confirm our data and investigate the possible role of LC-OCT for the different malignant skin tumours.

## Figures and Tables

**Figure 1 diagnostics-13-00361-f001:**
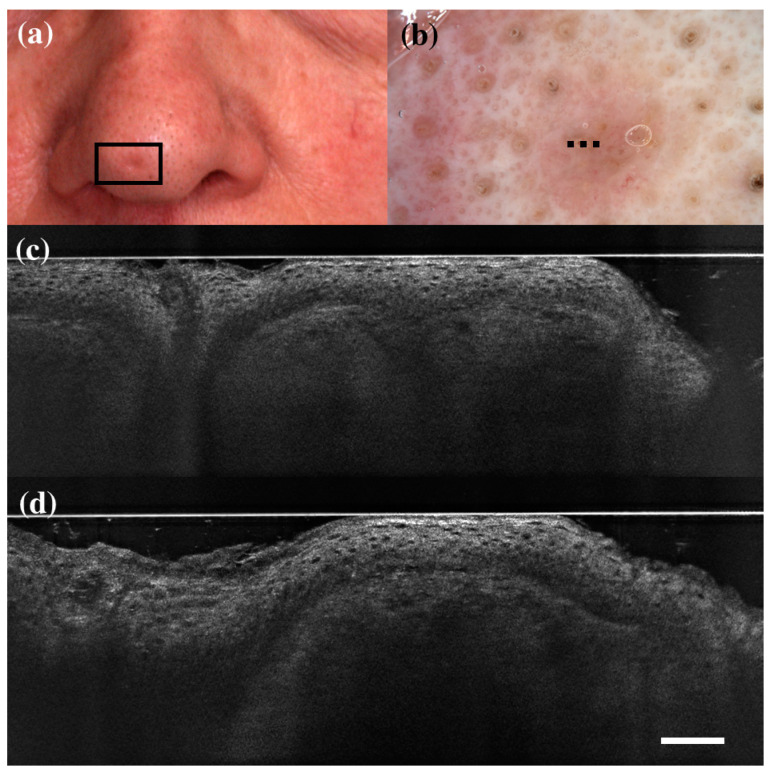
False positive case of basal cell carcinoma at dermoscopy. Clinical (**a**), dermoscopic (**b**), and LC-OCT images (**c**,**d**). Dermoscopy identified the lesion as a basal cell carcinoma, while LC-OCT as healthy skin. Histology confirmed the diagnosis of healthy skin. Dashed line (**b**) indicates the approximate area of the LC-OCT imaging. White scale bar in LC-OCT images: 100 µm.

**Figure 2 diagnostics-13-00361-f002:**
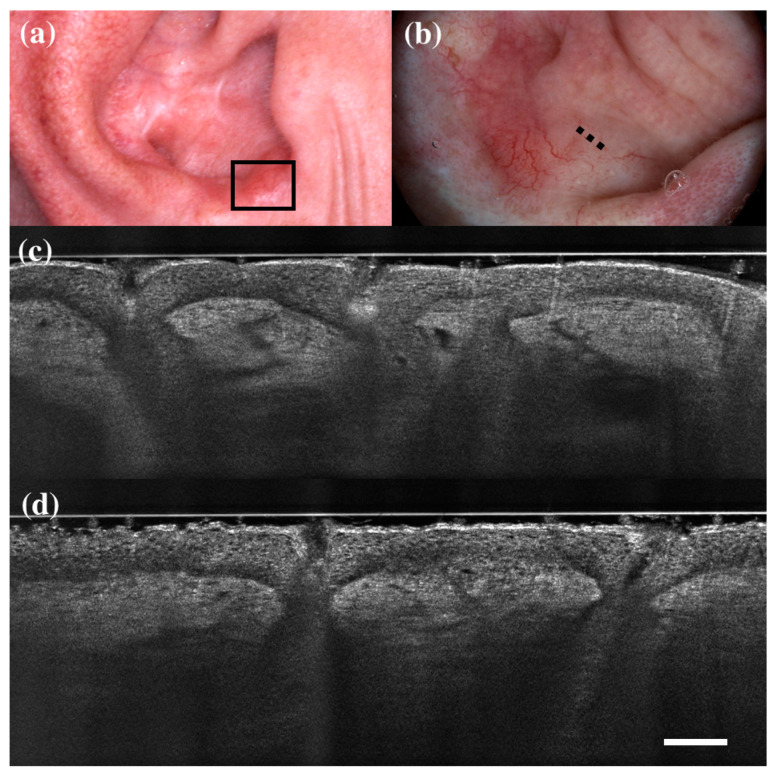
False positive case of basal cell carcinoma at dermoscopy. Clinical (**a**), dermoscopic (**b**), and LC-OCT images (**c**,**d**). Dermoscopy identified the lesion as a basal cell carcinoma, while LC-OCT identified it as healthy skin. Histology confirmed the diagnosis of healthy skin. Dashed line (**b**) indicates the approximate area of the LC-OCT imaging. White scale bar in LC-OCT images: 100 µm.

**Figure 3 diagnostics-13-00361-f003:**
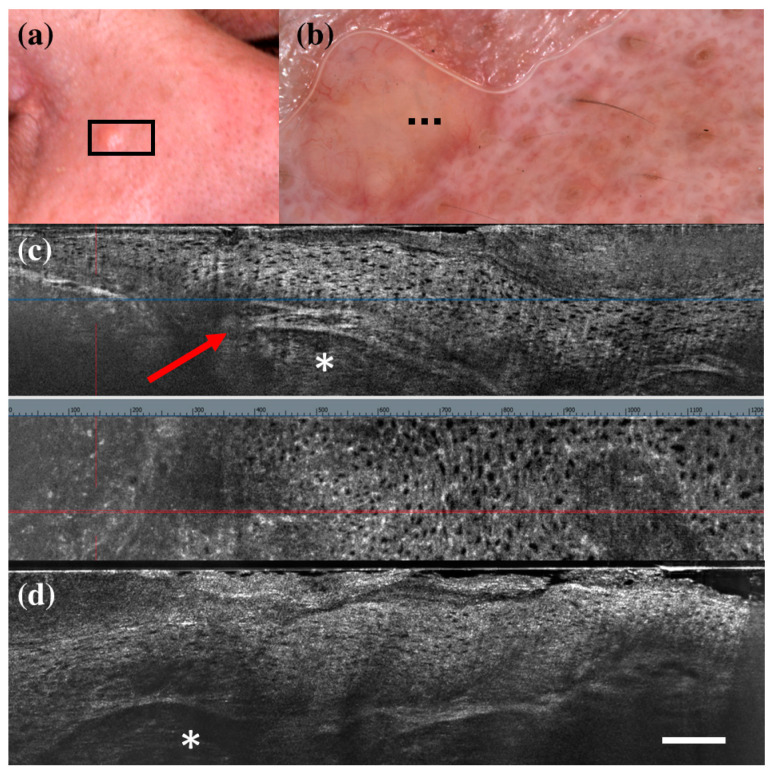
Sebaceoma diagnosed as basal cell carcinoma at dermoscopy and line-field confocal optical coherence tomography (LC-OCT). Clinical (**a**), dermoscopic (**b**), and LC-OCT images (**c**,**d**). Dermoscopy shows a pinkish-yellowish background and linear vessels. LC-OCT reveals a large lobular structure (asterisk) with “feuilletage” and clefting, surrounded by hyperreflective stroma and connected to a hair follicle (red arrow). Dashed line (**b**) indicates the approximate area of the LC-OCT imaging. White scale bar in LC-OCT images: 100 µm.

**Figure 4 diagnostics-13-00361-f004:**
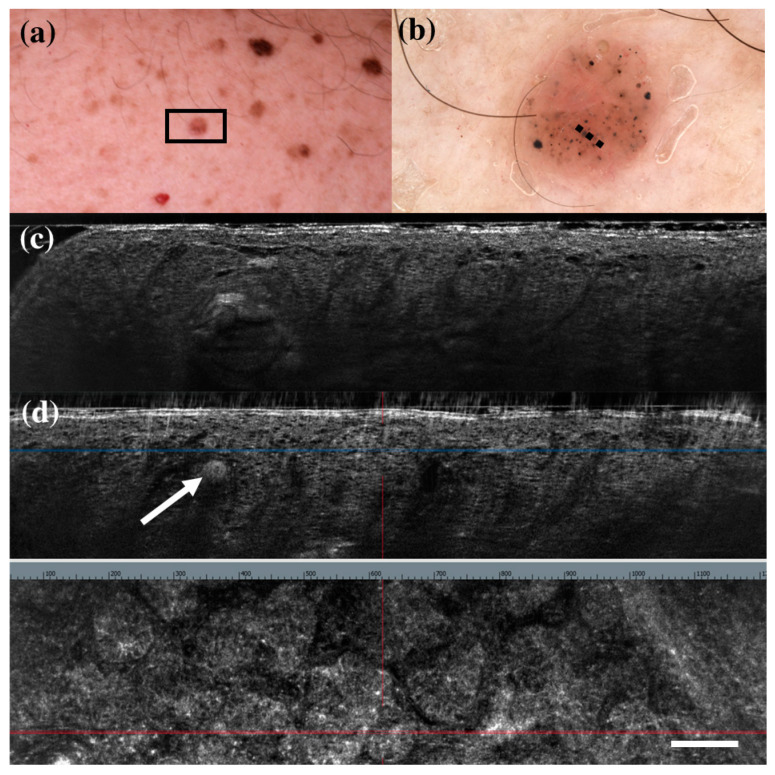
Trichoblastoma diagnosed as basal cell carcinoma under dermoscopy and line-field confocal optical coherence tomography (LC-OCT). Clinical (**a**), dermoscopic (**b**), and LC-OCT images (**c**,**d**). Dermoscopy shows black, brown, and grey globules and dots on a pinkish and brownish background. LC-OCT reveals small well-delimited roundish lobules with “palisading” and some keratin cysts (white arrow). Dashed line (**b**) indicates the approximate area of the LC-OCT imaging. White scale bar in LC-OCT images: 100 µm.

**Figure 5 diagnostics-13-00361-f005:**
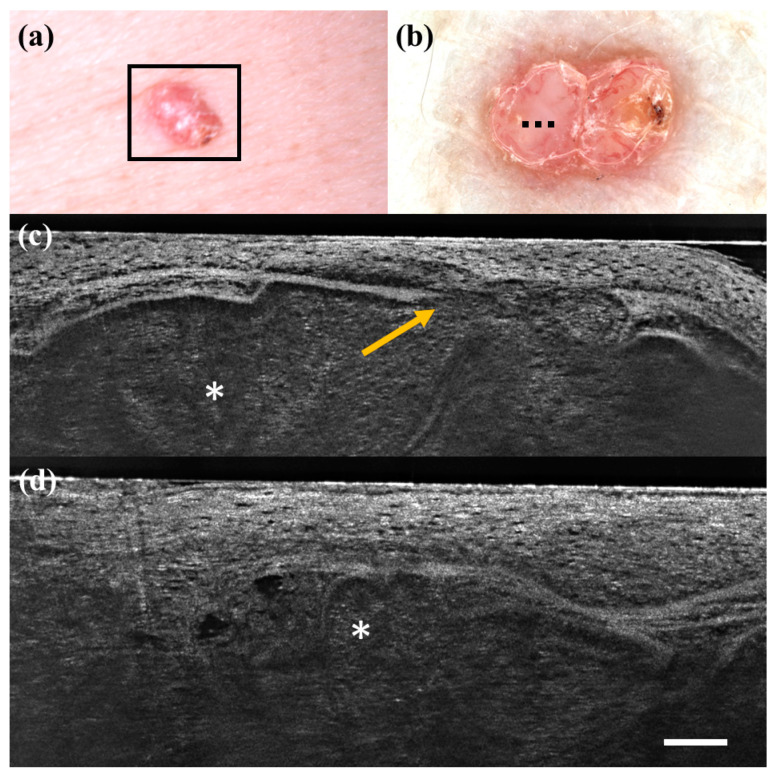
Trichillemoma diagnosed as basal cell carcinoma under dermoscopy and line-field confocal optical coherence tomography (LC-OCT). Clinical (**a**), dermoscopic (**b**), and LC-OCT images (**c**,**d**). Dermoscopy shows a pinkish background with linear vessels and scales. LC-OCT reveals lobular structures (asterisk) with clefting, palisade, “feuilletage”, and connection with the epidermis (orange arrow). Dashed line (**b**) indicates the approximate area of the LC-OCT imaging. White scale bar in LC-OCT images: 100 µm.

**Table 1 diagnostics-13-00361-t001:** Confusion matrix: dermoscopy vs. histology and LC OCT vs. histology.

		HISTOLOGY							
		BCC (n = 79)	Benign ML (n = 22)	Melanoma (n = 10)	AK (n = 16)	SCC (n = 22)	Inflammatory Lesion(n = 14)	Rare Disease (n = 5)	Other (n = 58)
**DERMOSCOPY** (in case of multiple diagnoses on dermoscopy, the worst diagnosis was retained)	**BCC (n = 96)**	76	2	0	1	2	3	0	12
	**Benign ML (n = 17)**	0	15	1	0	0	0	0	1
	**Melanoma (n = 14)**	0	5	9	0	0	0	0	0
	**AK (n = 15)**	1	0	0	12	1	0	0	1
	**SCC (n = 26)**	1	0	0	3	17	2	0	3
	**Inflammatory lesion (n = 6)**	0	0	0	0	0	5	0	1
	**Rare disease (n = 5)**	0	0	0	0	0	0	5	0
	**Other (n = 47)**	1	0	0	0	2	4	0	40
**LC-OCT** (in case of multiple diagnoses on LC-OCT, the worst diagnosis was retained)	**BCC (n = 84)**	77	1	0	0	0	1	0	5
	**benign ML (n = 20)**	0	17	1	0	0	0	0	2
	**Melanoma (n = 13)**	0	3	9	0	0	0	0	1
	**AK (n = 18)**	1	0	0	14	2	0	0	1
	**SCC (n = 24)**	1	0	0	1	19	1	0	2
	**Inflam (n = 9)**	0	0	0	0	0	8	0	1
	**rare disease (n = 5)**	0	0	0	0	0	0	5	0
	**Other (n = 53)**	0	1	0	1	1	4	0	46

**Table 2 diagnostics-13-00361-t002:** Sensitivity and specificity of dermoscopy and LC-OCT considering only histologically confirmed cases.

		DERMOSCOPY	LC-OCT	*p*-Value
**BCC** **(n = 79)**	**TP/P**	76/79	77/79	
	**TN/N**	127/147	140/147	
	**Sensitivity (CI)**	0.96 (0.89–0.99)	0.97 (0.91–1.00)	1
	**Specificity (CI)**	0.86 (0.80–0.91)	0.95 (0.90–0.98)	0.015
**SCC/Bowen** **(n = 19)**	**TP/P**	17/22	19/22	
	**TN/N**	195/204	199/204	
	**Sensitivity (CI)**	0.77 (0.55–0.92)	0.86 (0.65–0.97)	0.696
	**Specificity (CI)**	0.96 (0.92–0.98)	0.98 (0.94–0.99)	0.415
**AK/Bowen/SCC** **(n = 36)**	**TP/P**	33/38	36/38	
	**TN/N**	180/188	182/188	
	**Sensitivity (CI)**	0.87 (0.72–0.96)	0.95 (0.82–0.99)	0.428
	**Specificity (CI)**	0.96 (0.92–0.98)	0.97 (0.93–0.99)	0.785
**Malignant vs. non Malignant** **(n = 111)**	**TP/P**	105/111	106/111	
	**TN/N**	84/115	100/115	
	**Sensitivity (CI)**	0.95 (0.89–0.98)	0.95 (0.90–0.99)	1
	**Specificity (CI)**	0.73 (0.64–0.81)	0.87 (0.79–0.93)	0.013

TP, true positive; P, positive; TN, true negative; N, negative.

**Table 3 diagnostics-13-00361-t003:** Sensitivity and specificity of dermoscopy and LC-OCT considering histology and follow-up diagnoses.

		DERMOSCOPY	LC-OCT	*p* Values
**BCC** **(n = 79)**	**TP/P**	76/79	77/79	
	**TN/N**	127/160	153/160	
	**Sensitivity (CI)**	0.96 (0.89–0.99)	0.97 (0.91–1.00)	1
	**Specificity (CI)**	0.79 (0.72–0.85)	0.96 (0.91–0.98)	*p* < 0.001
**Malignant vs. non Malignant** **(n = 111)**	**TP/P**	105/111	106/111	
	**TN/N**	84/132	117/132	
	**Sensitivity (CI)**	0.95 (0.89–0.98)	0.95 (0.90–0.99)	1
	**Specificity (CI)**	0.64 (0.55–0.72)	0.89 (0.82–0.93)	*p* < 0.001

## Data Availability

Data are unavailable due to privacy restriction.

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
