# Peer review of "Diagnostic Accuracy of Line-Field Confocal Optical Coherence Tomography for the Diagnosis of Skin Carcinomas"

_diagnostics, 2023, doi:10.3390/diagnostics13030361_

Round 1

Reviewer 1 Report

The authors study the specificity and sensitivity of line-field confocal (LC-)OCT in comparison to conventional dermoscopy for detecting skin cancer. They investigated about 240 lesions and come to the conclusion that LC-OCT has a significantly higher specificity for diagnosing basal cell carcinomas than dermoscopy. When being applied after dermoscopic examination, it can improve the reliability of skin cancer detection and reduce the amount of unnecessary surgeries.

Before the manuscript can be considered for publication, the authors should take the following remarks into account:

The authors should discuss in more detail how skin cancer is detected from the LC-OCT images. How reliable are the results? As already stated by the authors, the interpretation is operator-dependent.

It would help if the authors could refer to the marked regions (asterisks, arrows) not only in the figure captions, but also in the main text.

The figures need some improvement. For better reference, the authors should mark in (a) the entire rectangular region that is displayed in (b). Similarly, they should mark in (b) the cross-sections measured with LC-OCT. Finally, all subfigures should contain scale bars. Some figures contain screenshots of scale bars, and horizontal blue/red lines (e.g. Fig. 3) which are barely visible. Also, the markers (especially the red arrow) are hard to see. The caption of Fig. 5 refers to a “green arrow” which is not shown in the figure.

While the text is mainly well written, there are some linguistic deficiencies. The authors should carefully check for typos and grammatical errors, e.g. “that can be used to diagnose skin cancer” (instead of “that ca be used to diagnosed skin cancers”), “to correctly diagnose 26 over 33 dermoscopic FP cases” (instead of “to correctly diagnosed…”), “that are already identified as suspicious” (instead of “that are already identify …”). There are also typos in all figure captions. There should be no punctuation before “(c-d)”, the phrase “while of LC-OCT” should be rephrased, and “line-filed” should read “line-field”. Please also check for other typos.

Author Response

Reply to reviewer 1

We added details on how skin cancer is detected from the LC-OCT images in the materials and method section, and we added in the discussion that an effort must go into defining objective criteria for the diagnosis of skin tumors to have more reliable and comparable results.

We referred to the marked regions (asterisks, arrows) not only in the figure captions, but also in the main text.

We improved the figures according to the reviewer’s suggestions. We marked in (a) the entire rectangular region that is displayed in (b). We marked in (b) the cross-sections measured with LC-OCT. We added scale bars. The horizontal blue/red lines are recorded with the images, and it is not possible to delete them. The markers are now thicker and better visible. In the caption of Fig. 5 “green arrow” has been changed into “asterisk”

We carefully check for typos and grammatical errors.

Reviewer 2 Report

The manuscript deal with the development of a new idea. The work has several issues that need to be addressed before considering it ready for publication.

A) The abstract is vague and seems to be incomplete. Issues such as which models were investigated? what was the mode of data collection? Are not clear to the readers.

B) The literature review section can benefit from the inclusion of works that are provided below;

C) The authors should shed light on the possibility of deep learning.

D) The conclusions should be presented in bullet format.

Author Response

Reply to reviewer 2

  1. The abstract has been changed following your suggestions. We perform a diagnostic study where we compared the diagnoses based on two different types of images provided by two different devices and we did not use any model.
  2. B) Unfortunately, we could not see your suggested references to be added.
  3. C) As suggested, we shed light on the possibility of deep learning in the discussion.
  4. D) The conclusions are presented in bullet format.
